# The Effect of Short-Term Heating on Photosynthetic Activity, Pigment Content, and Pro-/Antioxidant Balance of *A. thaliana* Phytochrome Mutants

**DOI:** 10.3390/plants12040867

**Published:** 2023-02-15

**Authors:** Vladimir D. Kreslavski, Alexandra Y. Khudyakova, Anatoly A. Kosobryukhov, Tamara I. Balakhnina, Galina N. Shirshikova, Hesham F. Alharby, Suleyman I. Allakhverdiev

**Affiliations:** 1Institute of Basic Biological Problems, Russian Academy of Sciences, Institutskaya Street 2, 142290 Pushchino, Russia; 2Department of Biological Sciences, Faculty of Science, King Abdulaziz University, Jeddah 21589, Saudi Arabia; 3K.A. Timiryazev Institute of Plant Physiology, Russian Academy of Sciences, Botanicheskaya Street 35, 127276 Moscow, Russia; 4Faculty of Engineering and Natural Sciences, Bahcesehir University, Istanbul 34353, Turkey

**Keywords:** *A. thaliana*, high temperatures, photosynthetic activity, oxidative stress, antioxidants, phytochrome mutants

## Abstract

The effects of heating (40 °C, 1 and 2 h) in dark and light conditions on the photosynthetic activity (photosynthesis rate and photosystem II activity), content of photosynthetic pigments, activity of antioxidant enzymes, content of thiobarbituric acid reactive substances (TBARs), and expression of a number of key genes of antioxidant enzymes and photosynthetic proteins were studied. It was shown that, in darkness, heating reduced CO_2_ gas exchange, photosystem II activity, and the content of photosynthetic pigments to a greater extent in the *phyB* mutant than in the wild type (WT). The content of TBARs increased only in the *phyB* mutant, which is apparently associated with a sharp increase in the total peroxidase activity in WT and its decrease in the *phyB* mutant, which is consistent with a noticeable decrease in photosynthetic activity and the content of photosynthetic pigments in the mutant. No differences were indicated in all heated samples under light. It is assumed that the resistance of the photosynthetic apparatus to a short-term elevated temperature depends on the content of PHYB active form and is probably determined by the effect of phytochrome on the content of low-molecular weight antioxidants and the activity of antioxidant enzymes.

## 1. Introduction

Heat stress induced by high environmental temperatures is seen as a major threat to agricultural production worldwide [1,2,3,4,5]. Elevated temperatures can cause oxidative stress induced with the production of reactive oxygen species (ROS), which may cause pigment and protein degradation, enzyme inactivation, damage to cell membranes, and inhibition of photosynthesis [2,6]. Photosynthesis is one of the key processes for plant productivity. However, it is the most heat sensitive, resulting in photosynthetic apparatus (PA) being damaged in the first line [6]. Therefore, it is important to know the PA targets and mechanisms of its protection under heat stress. Even moderate heating induces oxidative stress that leads to the formation of different ROS, damaging PA and other cell systems. Especially, the stress causes degradation of pigments, protein inactivation, and aggregation of various photosystem II (PSII) proteins [7]. On another hand, ROS such as H_2_O_2_ serve one of the early links in the stress signal network, which induces stress protection and acclimation [8], in which low-molecular antioxidants and antioxidant enzymes participate. Moreover, upon heating, the accumulation of heat shock proteins (HSPs) can play an important role in the acquired thermotolerance [9].

It is known that the joint action of moderate temperatures and light of a mild or moderate intensity does not cause serious damage to the PSII, but inhibits the repair of PSII, the activity of which is restored over time [6]. Note that the restoration of PSII activity is mainly associated with the synthesis of de novo proteins, in particular, PSII D1 and D2 proteins. The synthesis of these proteins is inhibited as a result of the formation of ROS [10], which leads to a decrease in the rate of photosynthesis and oxygen release. On the other hand, membrane-bound sensors seem to cause the accumulation of compatible solutes, such as glycine betaine, near PSII membranes. Moreover, heat stress induces the expression of genes for various protective proteins and enzymes, which facilitate the restoration of stress-damaged components of the PA and help its adaptation. However, key factors for recovery of the PA from heat-induced damage are moderate light and energy needed for the PSII restoration [6,10].

Heat and light are important natural factors that affect crop productivity; therefore, it is important to study their interaction [11,12]. Light is known to work by the photoreceptor network, to which belong receptors of blue-UV-A light (cryptochromes and phototropins) and red light (phytochromes (PHY)) [13]. It was indicated that the phytochromes that control many light-induced PA responses to stress factors [11,14,15] are also involved in the PA response to heat stress [11,12,16]. For example, PHYB is involved in Arabidopsis temperature perception and heat-tolerance induction [17]. It must be taken into account that only PHYB is assumed to be a temperature sensor in plants [18,19]. However, how phytochromes work in responses of the PA to elevated temperatures is unclear.

The joint effects of light and heat can be realized through the induction of the expression of different genes, which leads to the activation of the antioxidant system, enhancement of light-induced PA repair processes, induction of HSPs, changes in stomatal activity, and development of other protective mechanisms [17,20]. However, the molecular and physiological mechanisms that manifest themselves during short-term moderate heat stress in the PA, especially in the PSII, are not well understood, especially those associated with a role of phytochromes in ROS-induced restoration of the PSII and changes in pro-/antioxidant balance [11,17].

Thus, using *Arabidopsis* and tomato phytochrome-deficient mutants as an example, it was shown that these photoreceptors are involved in reactions to long-term [21] or short-term UV-B radiation [22], as well as to short-term exposure to high-intensity light [23] and to high temperatures mainly in long-term heating [11,12,17,18].

Considering an important role of light in heat-protection mechanisms, especially in PSII restoration, we studied the participation of phytochromes in responses of the PA to short-term heating. For this aim, *Arabidopsis* phytochrome mutants with deficits of key phytochromes PHYB and PHYA were used. We suggested that, in the development of stress-protection mechanisms, especially PSII restoration, the contents of the above phytochromes play an important role.

In the present work, the effects of short-term heat stress on the photosynthetic activity, pro-/antioxidant balance, content of photosynthetic pigments, and expression of a number of genes of *A. thaliana* mutants deficient in phytochromes B and A were studied.

## 2. Results

### 2.1. Plant Morphological Features

The mutant with a PHYB deficit had a longer hypocotyl than that of WT. The mutant leaves were also thinner compared with those of WT (by 10 ± 3%), as evaluated from weighing leaf cuttings of 7 mm in diameter. In addition, the mutant leaves were paler than those of WT. Here, *phyA* mutant characteristics were similar to those of WT, wherein heating for 2 h had an insignificant effect on these morphological parameters in the mutants.

### 2.2. Photosynthetic Activity

The effects of short-term heat stress (1 and 2 h, 40 °C) of mutants with a deficiency of PHYA and PHYB photoreceptors on fluorescent parameters characterizing the photochemical activity of photosystem 2 (PSII), such as the maximum quantum yield of PSII (F_v_/F_m_) and performance index (PI_ABS_), as well as the dissipation of the absorbed light energy into heat (DI_0_/RC), were investigated (Figure 1 and Figure 2). It was shown that, in plants taken out of the soil and heated in a cuvette with water for 2 h, the values of F_v_/F_m_ and PI_ABS_ significantly decrease, and the value of DI_0_/RC increases in both WT and mutants with a deficiency of PHYA and PHYB (Figure 1). The decrease in PSII activity and the increase in DI_0_/RC were the greatest with a PHYB deficiency. At the same time, the introduction of the protein synthesis inhibitor lincomycin (3 mM) significantly enhanced the decrease in F_v_/F_m_ and PI_ABS_ values during 2 h of heat stress only in the PHYB-deficient mutant (Figure 1), but not in WT. Moreover, DI_0_/RC increased in all variants under the action of temperature for 2 h.

With lincomycin, a significant increase in DI_0_/RC was observed only for the *phyB* mutant. At 1 h of heat exposure, a significant decrease in PSII activity (PI_ABS_) was observed only for the *phyB* mutant. Moreover, when keeping the mutants and WT at 40 °C in the dark, the decrease in PSII activity was more significant than in the light. Moreover, we observed similar trends for fluorescence parameters of plants placed in vessels with soil (Figure 2). The most significant changes in the parameters were also in the *phyB* mutant. After 6 h heat exposure under any conditions (dark or light) for any fluorescence parameters, the response of PSII to heating was the same in both WT and *phyB* mutants (Table 1).

Moreover, the rates of photosynthesis were similar. For example, under any light conditions before and after heating, the values of P_N_ were within 5–6 µmol CO_2_ m^−2^·s^−1^.

Plants subjected to heating at any time did not show any significant difference in fluorescent parameters when heated under light. However, when heated in the dark for 1 or 2 h, the fluorescent parameters were changed significantly and the changes were especially noticeable in the *phyB* mutant.

Initially, the rate of photosynthesis under the dark conditions differed little between WT and mutants (Figure 2). However, under the dark conditions, the rate of photosynthesis in WT when the plants were kept at 40 °C for 2 h decreased from 5.5 ± 0.3 μmol CO_2_ m^−2^·s^−1^ to 4.2 ± 0.1 μmol CO_2_ m^−2^·s^−1^; in the *phyA* mutant, it decreased from 6.5 ± 0.11 μmol CO_2_ m^−2^·s^−1^ to 4.7 ± 0.21 μmol CO_2_ m^−2^·s^−1^; whereas in the *phyB* mutant, it decreased from 5.6 ± 0.3 μmol CO_2_ m^−2^·s^−1^ to 2.4 ± 0.1 μmol CO_2_ m^−2^·s^−1^. At the same time, the respiration rate decreased only in the *phyB* mutant (from 2.4 ± 0.1 μmol CO_2_ m^−2^·s^−1^ to 1.4 ± 0.2 μmol CO_2_ m^−2^·s^−1^). Under light conditions, a significant decrease was observed only in the *phyB* mutant.

The transpiration rate (E_m_) in all variants changed little and was in the range of 1.7–2.0 mmol H_2_O m^−2^·s^−1^, except for *phyB* mutants heated in the light. In this case, the E_m_ value increased significantly from 1.7 ± 0.1 to 2.6 ± 0.2 mmol H_2_O m^−2^·s^−1^.

### 2.3. Pigments

Heat treatment of WT plants, under both dark and light conditions, did not noticeably affect the content of chlorophyll (Chl) (*a* + *b*) and carotenoids (Figure 3). However, in the dark, heating led to a decrease in the content of the pigments (Chl (*a* + *b*) and carotenoids) in mutants, but no significant difference was found under light conditions. Moreover, heating led to an increase in the content of ultraviolet-absorbing pigments (UVAPs) in all variants in both the light and the dark, and the increase was approximately the same in the mutants and WT. Thus, when the plants were heated in the dark, the content of photosynthetic pigments decreased in mutants; however, the most significant decrease in the content of the pigments was observed in the *phyB* mutant.

### 2.4. Pro-/Antioxidant Balance

After 2 h of heat treatment in the dark, a slight increase in the content of TBARs was found only in the *phyB* mutant (Figure 4). At the same time, the activity of GPX significantly increased only in WT (by 4.7 times), while in the mutant, the activity decreased. The increase in APX activity after heating was small for WT and greater for the mutant.

After heating the plants in the light, no noticeable difference was observed in the change in the content of TBARs in WT and the PHYB-deficient mutant. Moreover, only a slight increase in the activity of APX in WT and GPX in WT and this mutant was observed.

After 6 h dark exposure to heating, the TBARs content was reduced in WT and the *phyB* mutant. Conversely, activities of APX and GPX increased in both WT and the mutant (Figure 5). The greatest increase in APX activity was in the mutant and that in GPX activity was in WT.

### 2.5. Expression of Genes

Initially, in the *phyB* mutant, the levels of expression of *sAPX*, *APX1*, and *APX2* genes encoding antioxidant enzymes before heating were almost two times lower than in WT (Figure 6). When exposed to light, the level of expression of the *psbD* gene encoding the D2 protein was approximately two times lower in WT, while in the mutant, it changed slightly. On the other hand, the expression of the *tAPX* and *sAPX* genes was twice as high in WT, while no noticeable changes were found in the mutant under these conditions. In the dark, only the levels of expression of the *APX2*, *psbA*, and *psbD* genes in the *phyB* mutant were significantly lower than in WT.

## 3. Discussion

Global temperatures are predicted to rise in the future, leading to the requirement of more attention being paid to the study of the mechanisms of action of heat stress on plants and their PA. There are numerous data on targets and pathways for the recovery of damaged PA [6,20,24]. However, there are a number of gaps in this area. As a rule, we can observe in nature a combination of high temperature and exposure to sunlight. However, the effect of light irradiation is largely determined by the intensity of light and the state of photoreceptors such as phytochromes. However, the role of photoreceptors in the response of plants to stress caused by the combined action of both factors has been poorly studied.

Heat stress can lead to the accumulation of ROS in photosystems, especially in the PSII, as well as in the Calvin cycle. These ROS play a dual role, both by inhibiting photosynthesis and growth and damaging various plant systems by inducing the formation of various compounds that induce plant defense responses, in particular by inducing the synthesis of antioxidant enzymes (SOD, catalase, peroxidase, and some others) and low-molecular weight antioxidants (ascorbic acid, glutathione, tocopherols, carotenoids, and anthocyanins) [8,25], which play a significant role in ROS and lipid peroxidation products’ detoxification [26,27,28].

### 3.1. Heating under Dark Conditions

It has been shown that the rate of photosynthesis, PSII activity, and the content of photosynthetic pigments, especially in the case of PHYB deficiency, are the most sensitive to the negative short-term effect of elevated temperature when plants are kept in the dark, while PHYA deficiency does not cause such significant changes (Figure 2 and Figure 3). Moreover, we observed the greater increase in DI_0_/RC in the mutant than in WT. This parameter is an indicator of the dissipation of absorbed PSII energy when the photochemistry is reduced [29]. Indeed, upon heating, the PSII photochemical activity is reduced, resulting in the loss of absorbed energy and its transformation into heat, and the loss is higher in the *phyB* mutant. These results are consistent with the fact that the degree of oxidative stress, characterized by the content of TBARs after heating, significantly increased only in the *phyB* mutant, but not in the *phyA* mutant and WT (Figure 4). A simultaneous decrease in the level of TBARs and increase in the activity of both enzymes (APX and GPX) in the WT and *phyB* mutant after 6 h dark exposure to heating (Figure 5) can explain the similar changes in PSII photochemical activity of WT and the mutant upon heating.

It is assumed that PHYB is a key factor in the response of the PA to elevated temperatures [19]. This is consistent with our data showing that the negative effect of heating in the dark was the most pronounced in the *phyB* mutant. Apparently, this is due to the fact that, as a result of heating, the activity of total peroxidase significantly increased in WT, while, on the contrary, in the *phyB* mutant, it decreased.

It is also known that increased ROS formation under oxidative stress may be the result of inhibition of the activity of various antioxidant enzymes, such as catalase and peroxidase [25,30]. In particular, a decrease in peroxidase activity can contribute to the development of oxidative stress in mutant leaves. This may be due to the direct inactivation of enzymes by stressors (high temperature) or due to their binding to various intracellular intermediates [31]. The reduced level of expression of genes encoding various enzymes with peroxidase activity, which we found in the *phyB* mutant compared with WT, is also consistent with the fact that this activity was significantly lower in the mutant (Figure 4). Apparently, when heated for 2 h, enzymes with peroxidase activity have time to be synthesized, and this is a consequence of the increased formation of H_2_O_2_ in the initial period of heating. However, there may be an increase in peroxidase activity for other reasons [25,30,32]; for example, the activation of already synthesized enzymes [32]. Moreover, the increased PA resistance to a high temperature in WT compared with that in the *phyB* mutant may be due to the higher content of carotenoids in WT, which play the role of low-molecular weight antioxidants [33], as well as the elevated biosynthesis of HSPs [9].

Thus, when heated in the dark, the content of photosynthetic pigments decreases in mutants (Figure 3), while in WT, it changes slightly, which suggests that the phytochrome system controls the content of photosynthetic pigments. A lower accumulation of Chl in leaves at high temperatures may be associated with impaired Chl synthesis or its accelerated degradation, or both factors may affect the pigment content [34]. However, the phytochrome-controlled degradation of photosynthetic pigments rather takes place, as the synthesis of photosynthetic pigments is light-dependent. This is consistent with the data of [35], who found that Chl degradation is suppressed by short-term exposure of leaves to red light, and the effect of this light, in turn, was suppressed by far-red light. This result, according to the authors, indicates that phytochrome is involved in this process.

### 3.2. Heating under Light Conditions

It follows from our data that, under the joint action of heat treatment and light, there is no noticeable inhibition of PSII activity (Figure 1 and Figure 2), excluding P_N_ in the *phyB* mutant (Figure 2D). A lower efficiency in CO_2_ assimilation induced by the high temperature may be due to a decrease in the carboxylation activity of Rubisco, resulting in, for example, increased photorespiration [2] or the denaturation of Rubisco activase under heat stress [36]. However, the inhibition was significant, especially in the *phyB* mutant, in the presence of an inhibitor of protein synthesis, lincomycin; this is probably due to the fact that light-induced restoration of PSII in this mutant occurs more efficiently than in WT and the *phyA* mutant. This is consistent with the fact that under light conditions and high temperature, unlike in the darkness, phyB mutant did not demonstrate noticeable oxidative stress (Figure 4) inhibiting the synthesis of proteins necessary for the restoration of PSII activity. As a result, the PSII activity in the *phyB* mutant is restored quite effectively. The point is that moderate temperatures under the moderate light do not cause serious PSII damage, but inhibit the PSII repair [6]. The restoration of PSII activity is mainly associated with the synthesis of de novo proteins, in particular, PSII D1 and D2 proteins. The synthesis of these proteins is inhibited as a result of the formation of ROS, which decrease the rates of photosynthesis and oxygen release. In our case, in plants exposed to light, the level of expression of the *psbD* gene of the D2 protein was lower in WT, while in the mutant, it changed slightly (Figure 6). Such elevated expression of this gene can be one of the defense mechanisms and may explain the high effectivity of the restoration of the PSII in the mutant. Another defense mechanism that helps the mutant withstand stressful conditions when heated in the light may be its increased rate of water transpiration (E_m_). However, according to our data, this rate changed slightly. Another possible reason is that the small difference in photosynthetic activity between WT and the *phyB* mutant can be a more efficient formation of HSPs in the mutant than in WT. However, this is only one of the reasons, because, when protein synthesis is suppressed in both variants, there is a big difference between WT and the mutant. Moreover, in our case, induction of the antioxidant activity of enzymes or HSPs for a short time period (for example, 1 and 2 h heat exposure) is more likely than biosynthesis of low-molecular antioxidants. This agrees with similar induction of UVAPs in WT and the *phyB* mutant.

A deficit of PHYB leads to a lowered content of the PHYB active form, but after 2 h exposure of *phyB* plants to light, we found slightly lowered photosynthesis and, after 6 h the reduction was absent. Thus, we did not indicate an increase in resistance of the PA to short-term heating. In contrast to these data, the study of [11] showed that, with a longer exposure to stress (3 days), the degree of oxidative stress noticeably increased in tomato WT, but in PHYB1- (*tri*) and PHYB2- deficient (*phyB2*) mutants, stress was developed to a lesser extent. The authors suggested that mutation in PHYB genes triggered antioxidant responses to reduce the effects of ROS; however, the mechanism of this phenomenon is not known. Reductions in the photosynthetic rate and effective quantum yield of PSII and the apparent electron transport rate in tomato plants caused by a high temperature for some days were greater in *au* mutants with deficit of all phytochromes compared with WT [12]. Moreover, contents of Chl (*a* + *b*) and carotenoids in au leaves were lower those in WT. It is suggested that the deficit of all phytochromes in *au* mutants enhances heat-induced oxidative stress in these plants.

It seems that the action of phytochromes with heating can depend on the intensity and duration of heat stress as well as the state of the phytochrome system. For example, the authors of [16] showed that *Arabidopsis* thermoresistance may depend on the red/far-red light ratio, which affects the content of PHYB and its active form. The authors studied the action of supplementary far-red light (i.e., white light + far-red light compared with white light), taking into account that low red/far-red ratios reduce the proportion of PHYB in the active form. It was shown that the addition of far-red light enhanced the plant thermotolerance after heat shock (45 °C for 45 min) and the following recovery for 5 d under white light or white light + far-red light by lowering the content of PHYB in the active form. From another perspective, as mentioned above, with the long-term action of a high temperature (42 °C) [12], the deficit of all phytochromes led to the enhancement of the negative effect of heat stress on photosynthetic activity.

Thus, PHYB plays an important role in the resistance and adaptation of the PA and whole plants to elevated temperatures, but the manifestation of this effect mainly depends on light intensity and quality, the duration of temperature exposure, and other conditions.

## 4. Materials and Methods

### 4.1. Cultivation of Plants and Scheme of the Experiment

The plants *A. thaliana* of the ecotype Landsberg *erecta* (Ler) wild type (WT) and *phyA* and *phyB* mutants, deficient in PHYA and PHYB, respectively, were used in the experiments. Plant seeds were obtained from Nottingham Arabidopsis Stock Center (Nottingham, Great Britain). Plants were grown in pots with soil for 25 ± 1 days with a 12 h photoperiod at 23 ± 1 °C during the day and 21 ± 1 °C at night under white fluorescent lamps (I = 130 μmol (photon) m^−2^·s^−1^). Then, fully developed plants were subjected to short-term heat treatment (1, 2, 6 h; 40 °C). To obtain the desired temperature (40 °C), an electric dry-air thermostat TS-1/20 SPU (Smolensk, Russia) was used. Some of the plants were kept at a temperature in the dark and the other part was kept under white LEDs at an intensity of 45 µmol (photon) m^−2^·s^−1^. This weak light was used to repair heat-damaged PA.

In another experiment, plants carefully removed from the soil and placed in distilled water or in a solution of the inhibitor of protein synthesis lincomycin (3 mM) were used to separate PSII photodamage and light-induced restoration of this photosystem damaged by stress. The plants were kept in this solution for 2 h in the light at 40 °C and compared under the same conditions with plants without lincomycin. In this case, only fluorescent parameters were measured. In experiments with heating plants in vessels with soil, all physiological parameters listed below were measured.

For fluorescent and photosynthetic measurements, fully developed, healthy-looking upper leaves with nearly horizontal leaf blades were used. In each variant, 6–12 developed upper leaves from 3–4 plants were used. At least 10–15 leaves were used for each treatment to measure the pigment content and biomass. All experiments were repeated three or four times (*n*).

### 4.2. Photochemical Activity

Fluorescence parameters were assessed by the JIP-test using the fluorometer described in [37]. OJIP curves were measured under blue light illumination (455 nm, 5000 µmol (photon) m^−2^·s^−1^). The signal was recorded every 10 µs for 1 ms and every 1 ms from 1 ms to 1 s during data collection. The signal was transmitted from a silicon photodiode to a computer for further processing. The values F_0_, F_v_, and F_m_ were determined. Here, F_m_ and F_0_ are the maximum and minimum levels of Chl *a* fluorescence under dark adaptation conditions, respectively; F_v_ is the photoinduced change in fluorescence.

Based on the obtained induction curves, the F_v_/F_m_ values were calculated, DI_0_/RC is an indicator of the total energy dissipated by one active reaction center mainly in the form of heat, ABS/RC is the apparent size of the PSII antenna complex, and PI_ABS_ is the PSII performance index [29,38].

The following formula was used to calculate PI_ABS_:PI_ABS_ = (F_v_/F_m_)/(M_0_/V_J_) × (F_v_/F_0_) × ((1 − V_J_)/V_J_).(1)
where
M_0_ = 4 × (F_300µs_ − F_0_)/(F_m_ − F_0_)(2)
is the average value of the initial slope of the relative Chl *a* fluorescence variable, reflecting the rate of closing of PSII reaction centers, and
V_J_ = (F_2ms_ − F_0_)/(F_m_ − F_0_)(3)
is the relative level of fluorescence in phase J after 2 min.

### 4.3. CO_2_ Gas Exchange and Transpiration

The rates of photosynthesis, respiration, and transpiration were determined in a closed system under illumination using a portable infrared gas analyzer LCPro+ (ADC BioScientific Ltd., Hoddesdon, Great Britain, UK) at an ambient CO_2_ concentration of 420 ± 6.0 µmol m^−2^·s^−1^, an air temperature of 24–26 °C, and light intensity of 600 µmol (photon) m^−2^·s^−1^.

### 4.4. Photosynthetic and UV-Absorbing Pigments

The content of Chl *a* and *b* and carotenoids was determined in 96% ethanol extracts [39] by analyzing the absorption spectra of the samples on a Genesys 10 UV spectrophotometer (Thermo Fisher Scientific, Waltham, MA, USA) at λ_max_ of 470, 649, and 665 nm. The content of UVAPs was determined using fully developed, healthy-looking upper leaves (8–12), which were kept for 24 h in acid methanol (methanol/water/HCl, 78:20:2) at +4 °C [40]. The optical density of the samples was determined in the UV range (maximum at 327 nm) using a spectrophotometer (Genesys 10 UV, Thermo Fisher Scientific, Waltham, MA, USA). The content of UVAPs was expressed in relative units per 100 mg of dry matter.

### 4.5. Thiobarbituric Acid Reactive Substances and Enzyme Activity

The activity of ascorbate peroxidase (APX) was determined according to the method of [41] through the decrease in absorbance at 290 nm due to the oxidation of ascorbate. Guaiacol-dependent peroxidase (GPX) activity was measured spectrophotometrically as described [42] based on the conversion of guaiacol to its oxidized tetraguaiacol form and monitored at 470 nm. The content of products that react with thiobarbituric acid (TBARs) was determined according to the method described [42]. The absorption of TBARs was measured at 532 and 600 nm using a Hitachi-557 spectrophotometer (Hitachi Ltd., Kyoto, Japan). All data were calculated on 1 g wet weight basis.

### 4.6. Quantitative Real-Time PCR

Total RNA was isolated using TRI-Reagent (Sigma, St. Louis, MO, USA) according to the instructions supplied with the reagent. For the synthesis of the first cDNA strand, a reverse transcription kit (Sintol, Moscow, Russia) was used in accordance with the manufacturer’s instructions. Quantitative real-time PCR was performed using iCycler IQ5 (Bio-Rad, Alfred Nobel Drive, Hercules, CA, USA) and the reaction mixture from the qPCRmix-HS SYBR kit (Evrogen, Moscow, Russia). The *UBQ5* gene (AT3G62250) was used as an internal reference [43]. The *RPL2* gene (ATCG00830) was used as a control for the *psbD* gene. Primer sequences are presented in Table A1.

### 4.7. Statistics

Three biological and at least 6–12 analytical replicates were used for each experiment. Data are expressed as mean ± SD. One-way analysis of variance (ANOVA) with SigmaPlot 12.3 software (Systat Software Inc., San Jose, CA, USA) was used. Various letters were used to indicate significant differences between WT and mutants at *p* < 0.05.

## 5. Conclusions

It was found that the resistance of the *A. thaliana* photosynthetic apparatus to a short-term elevated temperature (1 or 2 h) depends on both the light intensity and the content of PHYB, probably in its active form, and to a lesser extent on PHYA. The resistance is apparently largely determined by the content of low-molecular antioxidants, HSPs and the activity of antioxidant enzymes, as well as the rate of light-induced recovery of damaged PSII. It is likely that the activity of PHYB depends on the ratio of the active form of phytochrome to its total pool, which is quite high if the irradiation spectrum has a high ratio of red/far-red light [15]. If the ratio of red/far-red light is low, under short-term moderate heating, the inhibition of photosynthesis is small. Under longer exposure to heat, PHYB deficiency had a weak influence on the PSII resistance to heat, even in dark conditions. We think that, after 6 h dark exposure to heat, the WT and *phyB* mutant chloroplast antioxidant potentials are moving close to each other. As a result, the heat resistance of the PSII in WT and the *phyB* mutant is the same.

It seems to be important in the future to compare the effects of deficiency and excess of PHYB during short-term and long-term heating.

## Figures and Tables

**Figure 1 plants-12-00867-f001:**
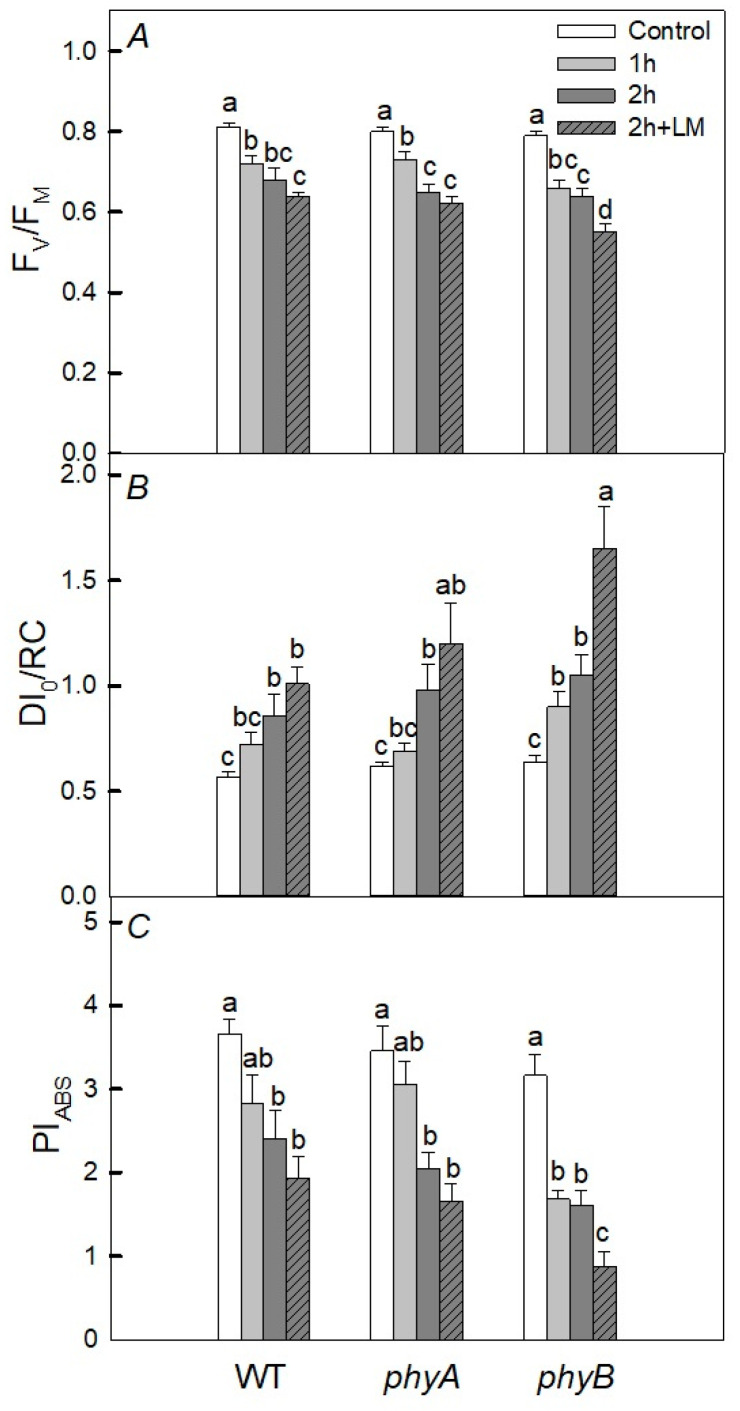
Effect of heat treatment on fluorescent parameters in 25-day wild type (WT) plants and *phyA* and *phyB* mutants grown under white fluorescent lamps (130 μmol (photon) m^−2^·s^−1^, photoperiod 12 h) at a temperature during the photoperiod of 23 °C and during the dark period of 21 °C. The plants were taken out of the soil and kept in distilled water for 1 and 2 h at a temperature of 40 °C under white LEDs (45 μmol (photon) m^−2^·s^−1^); some plants were kept in the solution of lincomycin (LM). Mean values ± SD are shown (*n* = 4). Different letters correspond to a significant difference in values at *p* < 0.05. Here, F_v_/F_m_ is the maximal quantum yield of PSII photochemistry (**A**), DI_0_/RC is the quantum yield of energy dissipation (**B**), and PI_ABS_ is the PSII performance index (**C**).

**Figure 2 plants-12-00867-f002:**
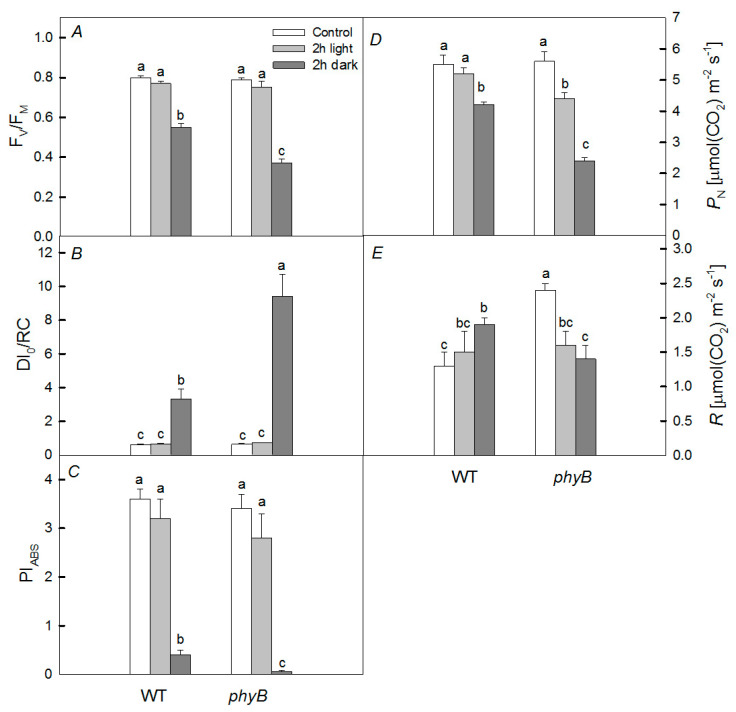
Effect of 2 h of heat treatment on fluorescent parameters (F_v_/F_m_ (**A**), DI_0_/RC (**B**), and PI_ABS_ (**C**)) and rates of photosynthesis (P_N_) (**D**) and respiration (R) (**E**) in wild type (WT) and *phyB* mutants. Plants were grown in tops and kept for 2 h at 40 °C in the dark (WT and *phyB*, dark) and in the light (WT, and *phyB*, light) compared with the initial variants at 21 °C. Mean values ± SD are shown (*n* = 4). Different letters correspond to a significant difference in values at *p* < 0.05.

**Figure 3 plants-12-00867-f003:**
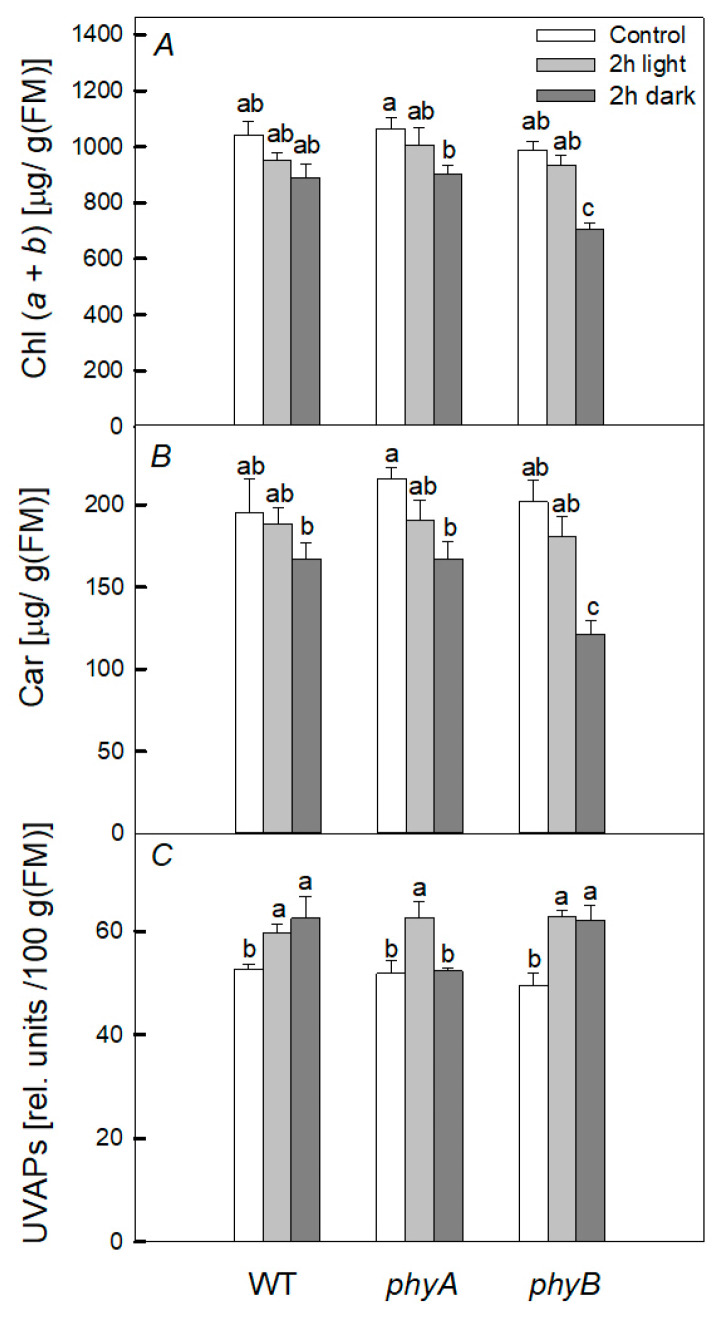
The effect of short-term heating on the content of Chl (*a* + *b*) (**A**) and carotenoids (Car) (**B**) and UV-absorbing pigments (UVAPs) (**C**) in WT and *phyA* and *phyB* mutants. The plants were kept in the dark (dark) and in the light (light) for 2 h at 40 °C. Mean values ± SD are shown (*n* = 4). Different letters correspond to a significant difference in values at *p* < 0.05.

**Figure 4 plants-12-00867-f004:**
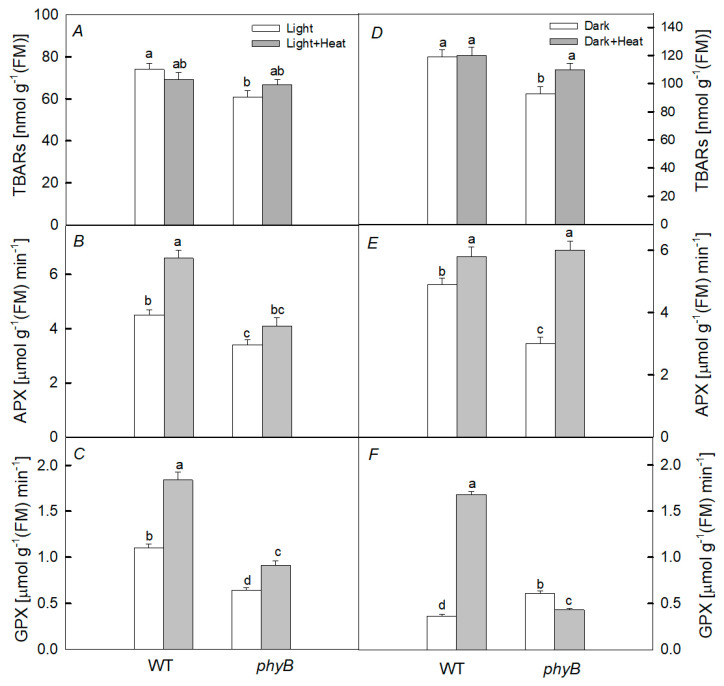
Effect of 2 h heat treatment 40 °C under light (**A**–**C**) and dark (**D**–**F**) conditions on the activity of the antioxidant enzymes ascorbate peroxidase (APX) (**B**,**E**) and guaiacol-dependent peroxidase (GPX) (**C**,**F**), as well as the content of thiobarbituric acid reactive substances (TBARs) (**A**,**D**) in the leaves of WT and *phyB* mutant plants. Mean values ± SD are shown (*n* = 4). Different letters correspond to a significant difference in values at *p* < 0.05.

**Figure 5 plants-12-00867-f005:**
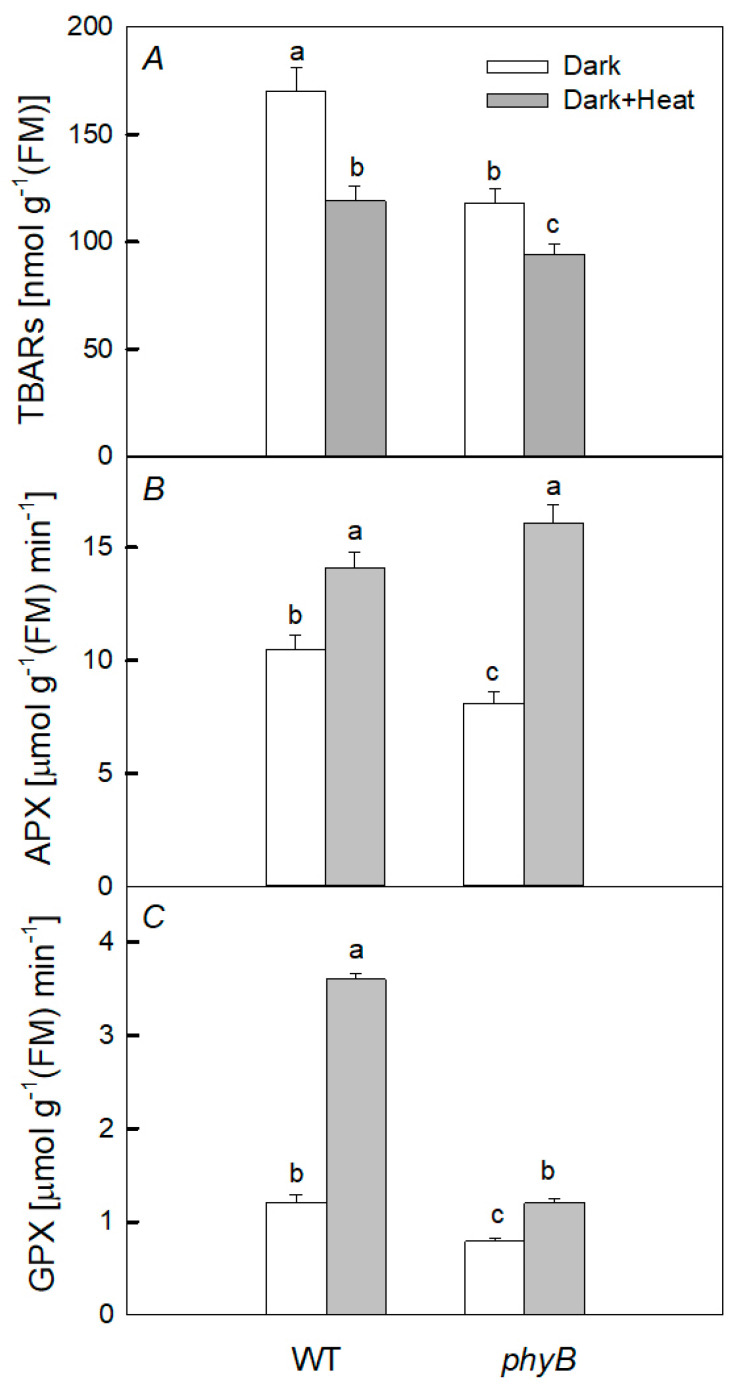
Effect of 6 h heat treatment (40 °C, dark) on the activity of the antioxidant enzymes ascorbate peroxidase (APX) (**B**) and guaiacol-dependent peroxidase (GPX) (**C**), as well as the content of thiobarbituric acid reactive substances (TBARs) (**A**) in the leaves of WT and *phyB* mutant plants. Mean values ± SD are shown (*n* = 4). Different letters correspond to a significant difference in values at *p* < 0.05.

**Figure 6 plants-12-00867-f006:**
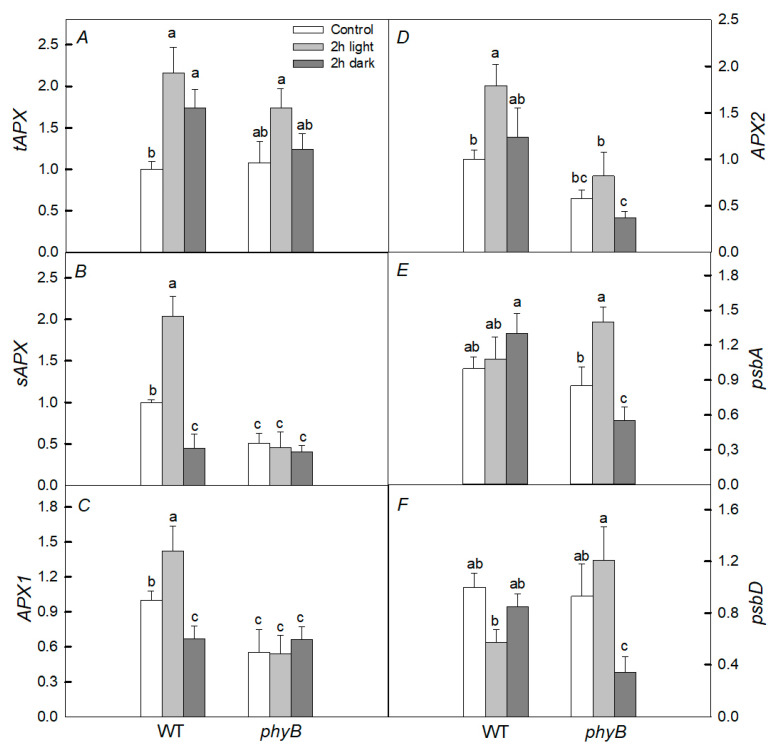
Effect of heat treatment (40 °C, 2 h) in the dark (dark) and in the light (light) on the expression of a number of *tAPX* (**A**), *sAPX* (**B**), *APX1* (**C**), *APX2* (**D**), *psbA* (**E**), and *psbD* (**F**) genes in WT plants and the *phyB* mutant. Mean values ± SD are shown (*n* = 3). Different letters correspond to a significant difference in values at *p* < 0.05.

**Table 1 plants-12-00867-t001:** Effect of heat treatment on fluorescent parameters in wild type (WT) and *phyB* mutants grown under white light (130 μmol (photon) m^−2^·s^−1^, photoperiod 12 h). Plants were grown in tops and kept for 6 h at 40 °C in the dark (WT, dark and *phyB*, dark) and in the light (WT, light and *phyB*, light) and compared to the initial variants at 21 °C. Mean values ± SD are shown (*n* = 4). Different letters correspond to a significant difference in values at *p* < 0.05. Here, F_v_/F_m_ is maximal quantum yield of PSII photochemistry, DI_0_/RC is the quantum yield of energy dissipation, and PI_ABS_ is the PSII performance index.

Parameters	DI_0_/RC	PI_ABS_	F_v_/F_m_
WT	0.57 ± 0.03 ^b^	3.66 ± 0.33 ^a^	0.78 ± 0.01 ^a^
WT, light	0.57 ± 0.03 ^b^	3.78 ± 0.39 ^a^	0.76 ± 0.01 ^a^
WT, dark	12.90 ± 3.00 ^a^	0.01 ± 0.01 ^b^	0.21 ± 0.03 ^b^
*phyB*	0.53 ± 0.03 ^b^	3.98 ± 0.39 ^a^	0.79 ± 0.01 ^a^
*phyB*, light	0.56 ± 0.04 ^b^	3.56 ± 0.31 ^a^	0.77 ± 0.01 ^a^
*phyB*, dark	12.50 ± 3.20 ^a^	0.01 ± 0.01 ^b^	0.21 ± 0.03 ^b^

## Data Availability

Not applicable.

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
