# Peer review of "The Effect of Short-Term Heating on Photosynthetic Activity, Pigment Content, and Pro-/Antioxidant Balance of A. thaliana Phytochrome Mutants"

_plants, 2023, doi:10.3390/plants12040867_

Round 1

Reviewer 1 Report

General comments

The article is quite interesting; however, there are some parts that need to be improved. The introduction part is difficult to follow and unclear. It must be restructured. The discussion part is a bit poor. It should be improved by including more international literature.

Specific comments:

Results:

2.2. Pigments

Line 150. This sentence “All figures and tables should be cited in the main test as Figure 1, Table 1, etc.” is a mistake?

Discussion

The discussion part is a bit poor. It should be improved by including more international literature. This section should not repeat results.

Bibliography and references in text should be checked.

Author Response

Answers to comments

First of all, we would like to thank the reviewers for their hard work in evaluating our MS.

Ref #1

General comments

The article is quite interesting; however, there are some parts that need to be improved. The introduction part is difficult to follow and unclear. It must be restructured. The discussion part is a bit poor. It should be improved by including more international literature.

Answer. We agree and improved the introduction part. This part has been completely rewritten, restructured and made clearer. Also, we improved the discussion part and it was completely changed. International literature was added (marked with yellow color in the list of Refs).

Specific comments:

Results:

2.2. Pigments

Line 150. This sentence “All figures and tables should be cited in the main test as Figure 1, Table 1, etc.” is a mistake?

Answer. Yes, this is mistake and the sentence is deleted.

Discussion

The discussion part is a bit poor. It should be improved by including more international literature. This section should not repeat results.

Answer. We agree. Discussion part was completely changed. We added a number of works on the topic of heat stress and phytochromes and discussed the results from these studies. 

Bibliography and references in text should be checked.

Answer. We checked bibliography and references in text of the MS.

Reviewer 2 Report

The manuscript is well collated and written well. A few concerns regarding this study.

1.     Short-term heating (2 h at 40 deg C): Does this bring heat stress response in the whole plant?

If yes, Authors have not provided any data such as heat shock factors A1a, HSP3, small subunit of Rubisco and ethylene response factor 1A to establish there was a heat shock response in the short-term heating in the whole plant.

2.     What is the physiological and morphological status of the heat-treated plants at the end of the plant growth cycle of different Arabidopsis genotypes?

3.     APX, together with MDHAR, dehydroascorbate reductase (DHAR), and GR, removes the H2O2. It would be nice to have the expression profile of these important genes as a part of the manuscript.

Author Response

Ref #2

The manuscript is well collated and written well. A few concerns regarding this study.

  1. Short-term heating (2 h at 40 deg C): Does this bring heat stress response in the whole plant?

Answer. If yes, Authors have not provided any data such as heat shock factors A1a, HSP3, small subunit of Rubisco and ethylene response factor 1A to establish there was a heat shock response in the short-term heating in the whole plant.

Answer. We partly agree. The fact is that our work was focused on the relationship of the phytochrome system with photosynthetic processes and changes in the general peroxidase activity and APX activity. Analysis of participation of HSPs, heat shock factors and other elements requires separate serious work. However, we took into account the comment and added corresponding parts to introduction and discussion sections where we discussed a role and possible participation of HSPs in indicated in our work effects.  Also, to emphasize that the focus of our work was placed more on photosynthetic processes and antioxidant balance and less on the analysis of gene expression we changed the title of the MS from: “The effect of short-term heating on photosynthetic activity, pigment content and pro-/antioxidant balance and gene expression of A. thaliana phytochrome mutants” to:  “The effect of short-term heating on photosynthetic activity, pigment content and pro-/antioxidant balance of A. thaliana phytochrome mutants.

  1. What is the physiological and morphological status of the heat-treated plants at the end of the plant growth cycle of different Arabidopsis genotypes?

Answer. We agree. We added to result section the part, which describes the phenotype of different Arabidopsis genotypes.

  1. APX, together with MDHAR, dehydroascorbate reductase (DHAR), and GR, removes the H2O2. It would be nice to have the expression profile of these important genes as a part of the manuscript.

Answer. We partly agree.  It would be good to add the expression profile of these important genes. However, it seems to us that data on general peroxidase activity characterize to a greater extent antioxidant potential than  the levels of enzyme transcripts since a direct relationship between levels of transcripts and activity of corresponding enzymes may not be. However, we will try to take into account this comment and analyze in the future the expression more an important for study expression of key genes during heating.

Ref #2

The manuscript is well collated and written well. A few concerns regarding this study.

  1. Short-term heating (2 h at 40 deg C): Does this bring heat stress response in the whole plant?

Answer. If yes, Authors have not provided any data such as heat shock factors A1a, HSP3, small subunit of Rubisco and ethylene response factor 1A to establish there was a heat shock response in the short-term heating in the whole plant.

Answer. We partly agree. The fact is that our work was focused on the relationship of the phytochrome system with photosynthetic processes and changes in the general peroxidase activity and APX activity. Analysis of participation of HSPs, heat shock factors and other elements requires separate serious work. However, we took into account the comment and added corresponding parts to introduction and discussion sections where we discussed a role and possible participation of HSPs in indicated in our work effects.  Also, to emphasize that the focus of our work was placed more on photosynthetic processes and antioxidant balance and less on the analysis of gene expression we changed the title of the MS from: “The effect of short-term heating on photosynthetic activity, pigment content and pro-/antioxidant balance and gene expression of A. thaliana phytochrome mutants” to:  “The effect of short-term heating on photosynthetic activity, pigment content and pro-/antioxidant balance of A. thaliana phytochrome mutants.

  1. What is the physiological and morphological status of the heat-treated plants at the end of the plant growth cycle of different Arabidopsis genotypes?

Answer. We agree. We added to result section the part, which describes the phenotype of different Arabidopsis genotypes.

  1. APX, together with MDHAR, dehydroascorbate reductase (DHAR), and GR, removes the H2O2. It would be nice to have the expression profile of these important genes as a part of the manuscript.

Answer. We partly agree.  It would be good to add the expression profile of these important genes. However, it seems to us that data on general peroxidase activity characterize to a greater extent antioxidant potential than  the levels of enzyme transcripts since a direct relationship between levels of transcripts and activity of corresponding enzymes may not be. However, we will try to take into account this comment and analyze in the future the expression more an important for study expression of key genes during heating.

Round 2

Reviewer 1 Report

The chances has been doing